# Two-Drug Regimens for HIV—Current Evidence, Research Gaps and Future Challenges

**DOI:** 10.3390/microorganisms10020433

**Published:** 2022-02-14

**Authors:** Alexandre Pérez-González, Inés Suárez-García, Antonio Ocampo, Eva Poveda

**Affiliations:** 1Group of Virology and Pathogenesis, Galicia Sur Health Research Institute (IIS Galicia Sur), Complexo Hospitalario Universitario de Vigo, SERGAS-UVigo, 36213 Vigo, Spain; eva.poveda.lopez@sergas.es; 2Infectious Diseases Unit, Department of Internal Medicine, Galicia Sur Health Research Institute (IIS Galicia Sur), Complexo Hospitalario Universitario de Vigo, SERGAS-UVigo, 36213 Vigo, Spain; antonioocampohermida@gmail.com; 3Infectious Diseases Group, Internal Medicine Department, Hospital Universitario Infanta Sofía, FIIB HUIS HHEN, 28703 San Sebastián de los Reyes, Spain; inessuarez@hotmail.com; 4Facultad de Ciencias Biomédicas y de la Salud, Universidad Europea, 28670 Madrid, Spain; 5CIBER de Enfermedades Infecciosas, 28029 Madrid, Spain

**Keywords:** human immunodeficiency virus (HIV), anti-retroviral treatment (ART), two-drug regimen (2DR), three-drug regimen (3DR)

## Abstract

During the last 30 years, antiretroviral treatment (ART) for human immunodeficiency virus (HIV) infection has been continuously evolving. Since 1996, three-drug regimens (3DR) have been standard-of-care for HIV treatment and are based on a protease inhibitor (PI) or a non-nucleoside reverse transcriptase inhibitor (NNRTI) plus two nucleoside reverse transcriptase inhibitors (NRTIs). The effectiveness of first-generation 3DRs allowed a dramatic increase in the life expectancy of HIV-infected patients, although it was associated with several side effects and ART-related toxicities. The development of novel two-drug regimens (2DRs) started in the mid-2000s in order to minimize side effects, reduce drug–drug interactions and improve treatment compliance. Several clinical trials compared 2DRs and 3DRs in treatment-naïve and treatment-experienced patients and showed the non-inferiority of 2DRs in terms of efficacy, which led to 2DRs being used as first-line treatment in several clinical scenarios, according to HIV clinical guidelines. In this review, we summarize the current evidence, research gaps and future prospects of 2DRs.

## 1. Introduction

Three-drug regimens (3DRs) became the main treatment for HIV infection in 1996, dramatically increasing the life expectancy of HIV-infected patients and lowering the number of acquired immunodeficiency syndrome (AIDS) events. However, 3DRs had some caveats: first, tolerability of first-generation HIV drugs was poor due to side effects and a high number of daily pills; secondly, drug–drug interactions were frequent and significant. In order to mitigate these factors while maintaining an adequate virological and immunological response, the development of 2DRs started around 2000. Most 2DRs tried to minimize ART-related toxicity by removing one or all NRTIs while maintaining a boosted PI as the main component of HIV treatment. Although less toxic than first-generation NRTIs, both abacavir (ABC) and tenofovir disoproxil fumarate (TDF) were associated with several side effects, including a higher frequency of non-AIDS events. An association between ABC and cardiovascular disease and an increased risk of myocardial infarction has been suggested [1]; however, to date, this relationship remains controversial [2] and the biological mechanisms that could link ABC and cardiovascular disease are not fully understood [3,4,5]. The STEAL clinical trial evaluated the switch from old NRTI regimens to a fixed-dose combination treatment including either abacavir/lamivudine (ABC/3TC) or emtricitabine/tenofovir disoproxil fumarate (FTC/TDF): after 48 weeks, no difference was found related to inflammatory biomarkers, insulin resistance or endothelial function [6]. On the other hand, there are conflicting results regarding the relationship between ABC and platelet activation, a biological mechanism related to cardiovascular disease [5]. The molecular mechanisms underlying these pro-coagulant scenarios associated with ABC have been extensively studied; however, to date, there is no conclusive explanation. Recently, higher levels of platelet-derived microvesicles (MVs) among patients receiving ABC/3TC than among those receiving TDF or TAF (tenofovir alafenamide)/3TC or FTC have been reported [7]. These findings might be linked to the increased platelet aggregation observed in patients on ABC/3TC compared to those on TAF [8]. However, these observations need confirmation in larger studies and their significance for the occurrence of HIV-related comorbidities needs to be explored. Whether the removal of ABC is related to a reversion in these inflammatory parameters remains unknown.

On the other hand, TDF is related to chronic kidney disease and bone demineralization [9]. Although rare, TDF can produce proximal tubulopathy, and some cases of Fanconi syndrome have been reported [10,11]. The novel tenofovir formulation, TAF, has lower renal and bone toxicity compared to TDF [12,13]. Moreover, substituting TDF for TAF as a backbone is related to an improvement in renal function (i.e., creatinine clearance) and bone mineral density [14], and a pooled analysis of 26 clinical trials comprising more than 9300 patients also reported a lower rate of proximal renal tubulopathy (0.34% vs. 0%) and discontinuations due to renal adverse events (0.47% vs. 0.05%) in patients who received TAF compared to TDF [15]. However, TAF seems to be associated with weight gain and metabolic changes (i.e., increase of low-density lipoprotein) [16,17,18].

The need for reducing toxicity, improving tolerability and the cost-effectiveness of ART motivated the development of several clinical trials searching for a 2DR with comparable efficacy to 3DR. Second-generation boosted PIs became the mainstay of many 2DR clinical trials due to their high genetic barrier for the development of drug resistance. Most of these studies included lamivudine (3TC), a second generation of NRTI with a good safety and efficacy profile, as the companion drug.

The second-generation boosted PI lopinavir/ritonavir (LPV/r), combined with 3TC, was the first drug that showed high efficacy in a 2DR [19,20]. The success of LPV/r was followed by next-generation PIs, atazanavir/ritonavir (ATV/r) [21,22] and darunavir/ritonavir (DRV/r) [23]. Moreover, integrase strand-transfer inhibitors (INSTIs) became available in 2007 and soon showed high efficacy, better tolerability compared to PIs and low potential for drug–drug interactions. Soon INSTIs were tested in 2DR and 3DR scenarios, both in naive and treatment-experienced patients. Raltegravir (RAL) was the first INSTI commercialized drug and replaced PIs in some clinical scenarios. In addition, RAL served as a companion drug for PIs in NRTI-sparing treatments [24]. Second-generation INSTI dolutegravir (DTG) combined with 3TC proved to be safe and effective in clinical trials among treatment-naïve and previously treated patients [25,26]. Since then, clinical evidence that supports the use of 2DRs has been increasing and most clinical guidelines have included 2DR as a preferred or alternative regimen both in naive and pre-treated HIV-infected patients [27,28,29] (Table 1 and Figure 1).

In this review, we summarize the current clinical evidence on 2DRs for the treatment of HIV infection, their role in selected clinical scenarios, their future as long-acting treatments and current research gaps.

## 2. Methods and Search Strategy

We searched in PubMed for publications related to clinical trials and observational studies, using a combination of the following terms and abbreviations: “HIV”, “two-drug regimen”, “three-drug regimen”, “naïve-to-ART”, “treatment experience”, “simplification”. We also consulted several clinical guidelines from different societies (GeSIDA, Madrid, Spain; DHHS, Washington, DC, USA; EACS, European Aids Clinical Society, Brussels, Belgium; BHIVA, Letchworth, UK). The literature search started in October 2021 and ended by January 2022 and included scientific data published between January 1996 and January 2022.

### 2.1. Efficacy

The first aim of 2DR was to demonstrate a similar efficacy compared to 3DR. The main concern was whether removing one drug from ART could cause a loss of virological suppression. Therefore, several clinical trials were designed in order to compare the virological efficacy of 2DRs and 3DRs. The main clinical trials evaluating 2DRs vs. 3DRs are summarized in Table 2.

#### 2.1.1. DR Based on Boosted Protease Inhibitors

Boosted PIs in combination with 3TC have been extensively tested in clinical trials. GARDEL [20] and OLE [19] were two open-label clinical trials comparing LPV/r plus 3TC and LPV/r-based 3DR. According to the intention-to-treat (ITT) analysis, virological suppression at week 48 was similar between the 2DR and its comparator 3DR both in GARDEL (88.3% vs. 83.7%, respectively) and OLE (88% vs. 87%, respectively) clinical trials. In addition, SALT and ATLAS-M [21,22] clinical trials compared switching to ATV/r plus 3TC versus continuing an ATV/r-based 3DR in virologically suppressed patients, achieving 2DR arm non-inferiority compared to the 3DR arm, according to ITT analysis. SECOND LINE was an open label, randomized clinical trial comparing LPV/r plus RAL versus LPV/r plus two or three NRTIs in patients who had confirmed virological failure. At week 48, the 2DR LPV/r plus RAL arm demonstrated non-inferiority versus its comparator (virological response according to ITT analysis 80.8% vs. 82.6%) [30]. The EARNEST and SELECT clinical trials had a similar design and showed consistent results [31,32].

Darunavir (DRV), a second-generation PI, has replaced both LPV and ATV in most ARTs, especially in high-income regions. The DUAL study was an open-label clinical trial comparing ritonavir-boosted DRV (DRV/r) plus 3TC versus DRV/r-based 3DR. The 2DR arm achieved non-inferiority compared to the DRV/r-based 3DR (the proportion of patients with virological suppression was 89% vs. 93%, respectively, according to ITT) [23]. The GeSIDA 9717 study performed a systematic review and meta-analysis of individual patient data evaluating PI-based 2DR clinical trials, including boosted LPV, ATV and DRV, showing the non-inferiority of 2DR [33].

DRV has also been tested in combination with other drugs, such as rilpivirine (RPV), a second-generation NNRTI. The PROBE-2 study was an open-label, non-inferiority clinical trial evaluating boosted DRV (bDRV) plus RPV as simplification therapy in virologically suppressed patients. At week 24, DRV plus RPV showed a similar virological response, achieving non-inferiority versus the control arm (90.0% vs. 93.8%, respectively) [34].

#### 2.1.2. Integrase Strand Transfer Inhibitor-Based 2DR

DTG has been tested in combination with several drugs. The double-blind clinical trials GEMINI-1 and GEMINI-2 compared DTG plus 3TC versus DTG plus FTC/TDF among ART-naive patients [25] with a HIV-RNA viral load under 500,000 copies/mL and a CD4 T-lymphocyte count above 200 cells/µL. Virological response rates were similar in the 2DR and control arms (91% vs. 93%, respectively). The TANGO clinical trial evaluated the efficacy and safety of switching from a TAF-containing 3DR to DTG/3TC. In this open label clinical trial, switching to DTG/3TC achieved a similar virological response to the control arm (93.2% vs. 93%, respectively) [26]. Furthermore, several real-life cohort studies have also found a great virological response in patients treated with DTG/3TC [35].

DTG plus RPV has been successfully tested in simplification scenarios. SWORD-1 and SWORD-2 were two open-label clinical trials evaluating simplification to DTG/RPV. Patients in the control arm continued with their 3DR, containing either a NNRTI (e.g., efavirenz), a PI (e.g., DRV/r) or an INSTI (e.g., RAL). At week 48, in a pooled analysis of SWORD-1 and -2, the DTG plus RPV arm showed a similar virological response compared to the control arms (95% vs. 95%, respectively) [36].

DTG has also been combined with boosted DRV (bDRV). The DUALIS study compared switching to DTG plus bDRV or to continue receiving a DTG-based 3DR among treatment-experienced patients for 48 weeks and reported a similar virological response in both arms [37]. In addition, several observational cohort studies have assessed 2DR DTG plus bDRV in real-life settings with favorable results, especially in pre-treated patients with limited ART options [38,39].

#### 2.1.3. Non-Nucleoside Reverse Transcriptase Inhibitor-Based 2DR

Two-drug regimens based on two NNRTIs have not been able to demonstrate non-inferiority compared to 3DR. The 2NN Study was an open-label clinical trial evaluating the combination of two first-generation NNRTIs, efavirenz (EFV) plus nevirapine (NVP) versus 3DR, in naive-to-treatment patients. The 2DR had an inferior virological response compared to both NPV and EFV-based 3DR [40].

### 2.2. Safety

One of the major goals for the 2DR development was to minimize ART toxicity by lowering the number of antiretroviral drugs. By avoiding TDF and ABC, it is believed that renal, bone and cardiovascular side effects could be reduced. In addition, a lower number of daily pills and a reduced risk of drug–drug interactions could reinforce adherence to ART and diminish the quantity or intensity of the side effects. The AEs reported in the main clinical trials comparing 2DR and 3DR are summarized in Table 3.

#### 2.2.1. DR Based on Boosted Protease Inhibitors

In the GARDEL clinical trial, grade 2–3 adverse events (AEs) possibly or probably drug-related were less frequent in the 2DR arm (30% vs. 44%, respectively). Moreover, safety events leading to discontinuation were less common in patients receiving dual therapy [20]. The OLE clinical trial found a mild improvement in renal function and a slightly worsening lipid profile in the 2DR arm [19]. The main cause of these changes was the removal of TDF, the most commonly withdrawn NRTI in the 2DR arm (62%). Overall, a similar rate of AEs was found in both arms of the SALT clinical trial, which showed a similar AE rate in the 2DR arm (70.7% vs. 70.2%, respectively), including AE leading to discontinuation [21]. The DUAL clinical trial also found a similar AE rate in the 2DR and 3DR arms [23]. On the whole, these clinical trials found a similar AE rate in PI-based 2DR arms and TDF was the most commonly removed drug (between 60 and 70% of patients in the 2DR arms). The withdrawal of TDF was related to a slightly worsening lipid profile but a mild, non-significant improvement in creatinine clearance, albeit the impact of these minor changes in clinical events remains unknown.

#### 2.2.2. Integrase Strand Transfer Inhibitor-Based 2DR

The GEMINI clinical trials also found a similar adverse event rate in 2DR and control arms (76% vs. 81% respectively) [25]. In addition, the 2DR arm was associated with better renal, urine and bone turnover biomarker profiles. TANGO, a clinical trial focused on removing TAF, found better lipid and renal biomarker profiles in the 2DR than in the control arm. Focusing on the SWORD clinical trial, adverse events were more common in the DTG/RPV arm compared to the control arm, especially neuropsychiatric events, attributed to DTG (77 % vs. 71%, respectively). Investigators imputed this difference to several factors, including the open-label design [26]. Similar results were found in the PROBE-2 clinical trial, with slightly more adverse events in the 2DR arm versus the control arm [34]. With regard to simplification clinical trials, patients who maintain a stable ART report fewer adverse events than those who change to a new ART.

The combination DTG/bDRV, tested in the DUALIS clinical trial, showed a similar adverse event rate at week 48 in the 2DR and control arms [37] On the other hand, the SECOND LINE clinical trial found a worsening lipid profile when switching to 2DR (RAL plus LPV/r). The increase in serum total cholesterol and low-density lipoprotein cholesterol (LDL) was probably related to the removal of TDF. However, the long-term relevance of these lipid changes remains unknown [30].

### 2.3. Emergence of Drug-Resistance Mutations

The emergence of treatment-associated resistance mutations (RAM) was a major concern at the beginning of the development of 2DRs, especially regarding NRTIs, for which a single mutation can compromise their treatment efficacy, as in the case of the M184V mutation associated with 3TC or FTC resistance. The first 2DRs were designed for pre-treated patients who had a suppressed viral load at the time of study recruitment. Emergence of RAM was very uncommon in PI-based 2DRs. During the OLE clinical trial, one patient randomized to the 2DR arm developed 3TC resistance with no cases in the 3DR arm. In the GARDEL clinical trial, the M184V mutation was present at treatment failure in two patients randomized to the 2DR arm but not within the 3DR arm. No mutations associated with PI resistance were found in either group. In the SALT clinical trial, the M184V mutation was detected in one patient randomized to the 3DR. No RAMs related to ATV were detected in either arm. In the DUALIS clinical trial, no RAMs related to 3TC were detected in either arm. One patient randomized to the 3DR arm developed mutations at the protease during virological failure, but the virus remained fully susceptible to DRV. A few years later, the GEMINI 1 and 2 trials observed similar results and no difference in the emergence of RAMs was found between the 2DR and 3DR arms [25]. Ten patients met criteria of virological failure through week 48, six in the 2DR arm and four in the 3DR arm. One patient maintained virological suppression until week 128, albeit an elevated HIV-RNA VL was detected at week 132. At week 140, HIV-RNA returned to levels under 50 copies/mL, but subsequently low-level viremia was detected at week 144 followed by withdrawal from the study due to lack of efficacy. A HIV genotypic-resistance test was performed in samples from weeks 132 and 144; the presence of the M184V mutation was identified at week 132 and the INSTI resistance mutation R263R/K at week 144, the latter confers a 1.8-fold change in susceptibility to DTG [41]. In these clinical trials, randomization was stratified in two groups depending on viral load. The HIV-RNA threshold was established in 100,000 copies/mL and no differences were identified between 2DR and 3DR in both groups. However, both trials have only recruited ART-naïve patients with a viral load lower than 500,000 copies/mL in the screening phase and only a small number of patients had a HIV-RNA higher than 500,000 copies/mL at the time of ART initiation, 13 in the 2DR arm and 15 in the 3DR arm. In regard to the emergence of RAMs, no difference was observed in this sub-group, but the sample size was too small, and no robust conclusions can be drawn.

The ART-PRO study was an open-label, proof-of-concept clinical trial evaluating DTG plus 3TC in patients for whom the M184V mutation associated with 3TC/FTC resistance was previously detected. The results demonstrated a similar virological response between patients with and without the M184V mutation [42]. The ANRS 167 LAMIDOL clinical trial evaluated DTG plus 3TC as simplification therapy in patients with suppressed viremia for 48 weeks and no RAMs were reported during the study period [43]. In addition, SOLAR 3D is an ongoing, open-label clinical trial evaluating the switch to DTG plus 3TC from a stable two-, three- or four-drug regimen. Patients were assigned to one of two arms, based on the prior detection of the M184V/I mutation. According to ITT analysis at week 48, there were no differences in viral suppression between both arms (HIV-RNA < 50 copies/mL 92% vs. 88%, respectively) [44].

On the other hand, the ACTG A5262 evaluated DRV/r plus RAL in ART-naïve patients. In this open-label, single-arm clinical trial, a basal VL higher than 100,000 copies/mL was strongly associated with virological failure and the emergence of RAL RAM [24]. During the NEAT 001/ANRS 143 clinical trial which evaluated DRV/r plus RAL vs. DRV/r plus FTC/TDF in ART-naive patients, both the VF and the emergence of RAM were strongly associated with having a baseline VL higher than 100,000 copies/mL. However, only RAL was related with the emergence of RAMs, not DRV [45].

### 2.4. Inflammation and Low-Level Viremia

There are some concerns about the impact of 2DR in several outcomes, such as low levels of HIV replication and inflammation. Despite an effective ART, low-grade HIV replication could persist in lymphatic tissues [46,47]. Persistent inflammation and immune activation play a key role in the development of non-AIDs events (i.e., cardiovascular disease or cancer) [48,49]. Indeed, some studies suggest that despite ART, HIV might interfere with cell–cell communication, increasing the release of several microvesicles, promoting viral replication and the liberation of proinflammatory cytokines [7].

#### 2.4.1. Inflammatory Biomarkers

The main concern is whether 2DRs are as effective as 3DRs at controlling inflammation and immune activation. A sub-study of the TANGO clinical trial evaluated several biomarkers, including d-dimer, high-sensitivity C reactive protein (hsCRP), interleukin-6 (IL-6), soluble CD14 (sCD14) and soluble CD163 (sCD163). The switch to DTG/3TC was related to a slight increase in IL-6 and sCD14 levels, but d-dimer, hsCRP and sCD163 levels remained unchanged at week 48 [26]. A sub-study of the NEAT001/ARNS143 compared the evolution of inflammatory biomarkers between DRV/r plus RAL vs. DRV/r plus FTC/TDF. At week 48, no difference was found between both arms regarding the concentrations of interleukin-1b (IL-1b), IL-6 and tumor necrosis factor alpha (TNF-α) [50]. The TDF arm was also associated with worse bone biomarker profiles (i.e., type 1 C terminal collagen crosslinks, urine CTX-1/creatinine, osteoprotegerin).

An Italian randomized clinical trial evaluated if the switch from a DTG-based 2DR to elvitegravir/cobicistat/FTC/TAF was associated with a decrease in residual viremia, defined as a detectable HIV-RNA below 50 copies/mL. At week 48, there were no differences between both arms regarding residual viremia and inflammatory biomarkers (i.e., d-dimer, C reactive protein, CD4/CD8 ratio) [51]. On the other hand, an Italian real-life cohort showed an increase in CD8 lymphocyte count when switching from 3DR to 2DR: this rise in the CD8 T lymphocyte count may be signaling a persistent immune activation [52].

#### 2.4.2. Low-Level HIV Replication

Several biomarkers have been proposed as HIV replication predictors, such as circulating HIV-DNA [46]. ART reduces HIV-DNA levels among other replication biomarkers, but low-level HIV replication remains active [53]. Some studies have searched for differences between low-level replication biomarkers in 2DR and 3DR patients. A sub-analysis of the AtLAS-M trial focused on circulating HIV-DNA levels at baseline and during follow-up. HIV-DNA levels were similar between the 2DR and 3DR arms at baseline. Nadir CD4 T lymphocyte count was inversely correlated with baseline HIV-DNA but no relation was found between HIV-DNA levels and CD4 T lymphocyte count at baseline. A decrease in HIV-DNA level was detected through week 48, with no difference between the 2DR and 3DR arms [54]. In addition, low level viremia (LLV) has emerged as another biomarker of HIV persistent replication although its definition varies across studies due to differences in RNA-HIV quantification methods. Previous studies have reported an association between LLV, defined as an HIV-RNA VL between 1 and 20 copies/mL, higher levels of pro-inflammatory biomarkers (i.e., d-dimer, soluble CD163, soluble endothelial protein C receptor) [55,56], a larger HIV latent reservoir size [57] and a higher risk for non-AIDS events or death [58]. Moreover, contradictory results have been reported regarding whether LLV may be different between PI-based and DTG-based 3DRs [57,59,60,61]. Recently, the GEMINI 1 and 2 clinical trials reported no differences regarding LLV between the 2DR and the 3DR arms [62].

### 2.5. Special Scenarios

Although evidence of 2DR efficacy and safety has dramatically increased in the past few years, there are some scenarios which deserve special attention.

#### 2.5.1. Severe Immunosuppression

Efficacy of 2DR in severe immunosuppressed patients remains unknown to this date. Most of the clinical trials have excluded severe immunocompromised patients, both in 3DR and 2DR. The NEAT001/ANRS143 clinical trial evaluated DRV/r plus RAL vs. DRV/r plus FTC/TDF in naïve-to-ART patients, including a sub-group of patients with CD4 lymphocyte counts lower than 200 cells per µL. In this sub-group, DRV/r plus RAL was inferior to DRV/r plus FTC/TDF [63]. The EARNEST clinical trial included 787 patients with a CD4+ lymphocyte count under 100 cells/µL in three arms: LPV/r plus two NRTIs, RAL plus LPV/r and LPV/r monotherapy. Regarding virological response, no differences were found between the 3DR and 2DR arms [32].

#### 2.5.2. HIV Viral Load above 500,000 Copies/mL

The GEMINI 1 and 2 studies are the largest clinical trials evaluating 2DR in ART-naïve patients. In both clinical trials, participants with a VL above 500,000 copies/mL during the screening visit were excluded. Only a few participants (*n* = 28) had a VL over this threshold at baseline and all continued to the randomization phase. Although effectiveness was similar in the 2DR and 3DR arms, it is important to note that the number of patients with very high viral loads was too small and no solid conclusions can be drawn from these results [25]. However, this limitation can be extended to 3DRs, since most clinical trials excluded those patients with high VL. The NEAT 001/ANRS 143 clinical trial found a higher risk of virological failure in patients with a VL over 100,000 copies/mL [45].

#### 2.5.3. Test-and-Treat Scenarios

During the last years, the test-and-treat strategy was developed in order to increase the ART initiation rate and to improve the linkage to healthcare systems. Briefly, test-and-treat seeks to initiate ART on the same day as HIV diagnosis, aiming for a rapid decline in HIV-RNA viral load and a decrease in the risk for HIV transmission. As the treatment is initiated on the same day of HIV diagnosis, much clinical information is not available when choosing an ART regimen (i.e., genotypic or phenotypic HIV-1 resistance test, HIV-RNA viral load, HBV serology, CD4-lymphocite count). In order to minimize the risk of virological failure, the ART for test-and-treat must have a high genetic barrier and be suitable for patients with a high HIV-RNA viral load and a low CD4 lymphocyte count. Moreover, if HBV chronic infection cannot be ruled out prior to ART initiation, HIV treatment must contain two drugs with HBV activity, especially in geographical areas with a high prevalence of HBV chronic infection. Test-and-treat strategies have been successfully tested in some low-income countries in Africa and several healthcare facilities across the United States [64,65]. During the early stages of the development of the test-and-treat strategy, concerns arose whether not having a resistance test could increase the risk for virological failure if one drug was not fully active against HIV. However, the prevalence of baseline HIV mutations compromising INSTI efficacy is very low [66,67]. In addition, baseline M184V mutation is very uncommon, according to several studies [25,68,69,70]. The STAT clinical trial is a 52-week pilot study evaluating DTG plus 3TC in test-and-treat scenarios without prior genotypic or phenotypic HIV-resistance test, HBV serology, HIV RNA viral load or CD4 lymphocyte count. According to ITT analysis, 78% participants achieved HIV-RNA less than 50 copies/mL [71]. To date, no clinical trials have compared 3DRs and 2DRs in test-and-treat settings, although evidence suggests that both options could be suitable for most patients.

### 2.6. Novel Formulations

In the past few years, new long-acting regimens have been developed as a combination of two subcutaneous or intramuscular-formulated drugs. Currently, there are several clinical trials evaluating these new 2DR long-acting treatments both in naïve-to-ART and treatment-experienced patients.

#### 2.6.1. Clinical Trials in Naïve-to-ART Patients

The first tested intramuscular combination included cabotegravir (CBG), an INSTI with a similar structure to DTG, and a novel formulation of RPV. The FLAIR study was a clinical trial evaluating intramuscular CGB/RPV in naïve-to-ART patients. After an induction oral phase with DTG/3TC/ABC for 20 weeks, patients were randomized to switch to CGB/RPV or to continue receiving oral DTG/3TC/ABC. Those patients randomized to the experimental arm completed a second induction phase based on oral CGB/RPV for a month and then switched to intramuscular monthly administration. At week 48, virological suppression was non-inferior compared to the experimental arm, according to ITT analysis [39]. LATTE-2 also included naïve-to-ART patients, who received an oral induction phase based on CBG/3TC/ABC for 20 weeks, and those virologically suppressed were randomized to one of the three following arms: intramuscular CBG/RPV every 4 weeks, intramuscular CBG/RPV every 8 weeks or oral CBG/3TC/ABC. At week 96, virological response was similar across the three arms (87% vs. 94% vs. 84% for 4 weeks administration, 8 weeks administration and control arms, respectively) [72]. A recent sub-study of the FLAIR clinical trial included 111 patients who transitioned to the injection group without an oral induction phase, compared to 121 patients who completed the oral induction. At week 124, the virological suppression rate was similar within both arms (99% vs. 93%, respectively), and adverse events leading to discontinuation were uncommon (less than 1% in both arms) [73].

#### 2.6.2. Clinical Trials in Treatment-Experienced Patients

ATLAS-2M is an ongoing clinical trial evaluating the switch to intramuscular CBG/RPV every 4 or 8 weeks in virologically suppressed patients. Virological response was similar between both arms at week 48, according to intention-to-treat exposed analysis [74]. A second long-acting ART based on islatravir (ISL) and doravirine (DOR) is currently under investigation. ISL is a first-in-class nucleoside reverse transcriptase translocation inhibitor (NRTTI), while DOR is a next-generation NNRTI. In contrast to CBG/RPV, the administration of ISL plus DOR is subcutaneous, potentially allowing a future implant device [75]. A phase II clinical trial evaluated oral ISL/DOR in virologically suppressed patients after an induction phase of ISL/DOR and 3TC or TDF for 24 weeks according to randomization. Patients receiving ISL were randomly assigned to three different doses: 0.25 mg, 0.75 mg and 2.25 mg. Virological response at week 48 was similar in the ISL arms (0.25 mg, 0.75 mg and 2.25 mg) and control (90% vs. 90% vs. 87% vs. 87%, respectively) [76].

### 2.7. Remaining Questions and Research Gap Areas

The efficacy and safety of 2DRs have been successfully tested in many clinical trials and real-life studies; several guidelines have included 2DRs as recommended regimens in treatment-naïve and treatment-experienced subjects (Table 3). However, there are some remaining questions. First, it is unknown whether 2DRs may be less effective than 3DRs in controlling immune activation. Some clinical studies have found no differences in cytokine production and levels of inflammatory biomarkers between 2DRs and 3DRs [26], but only a few trials have evaluated HIV-DNA and other viral reservoir indicators [54]. Another concern is the relationship between immune activation and non-AIDS events (i.e., cardiovascular disease, cancer). Previous research has found a causal relation between HIV replication, persistent immune activation and chronic diseases [48]. To date, clinical trials focused on inflammatory biomarkers have been conducted for short periods of time (i.e., 48 or 96 weeks) and there is some concern regarding whether the evolution of these parameters may influence the clinical outcomes in PLWH and whether a 2DR strategy could be associated with a poorer long-term prognosis compared to 3DRs (i.e., a higher incidence of non-AIDs events, such as cancer or stroke). Long-sighted studies focused on non-AIDS events incidence should be conducted in patients receiving 2DRs.

Second, it is also unknown whether 2DRs may have a worse diffusion to some anatomic locations, such as the central nervous system (CNS). Despite an effective ART, HIV-RNA could be detected at low levels in the cerebrospinal fluid (CSF) [77], although its clinical relevance is not fully understood. Previous studies have reported higher levels of several inflammatory biomarkers (e.g., IL-6, interferon gamma) in PLWH diagnosed with cognitive impairment [78]. The penetration of each drug in CSF varies widely based on several circumstances: ART adherence, drug–drug interactions, hematoencephalic barrier disruptions, etc. DTG exceeds the in vitro 50% inhibitory concentration (IC50) in CSF and shows a robust activity against HIV in the CNS [79,80], as well as RPV [81,82], 3TC [83] and DRV/r [84]. Some authors suggest that removing one component from ART could lead to a significant decrease of drug concentrations in these locations [85], albeit increasing evidence suggests that HIV-RNA suppression is non-inferior in 2DRs compared to 3DRs [86].

Pregnancy is another remaining question. To date, 3DR continues to be the main treatment for HIV-infected pregnant patients [87]. There are no clinical trials regarding 2DRs in pregnant women. In addition, ART prescription during pregnancy is conditioned due to pharmacokinetic issues (e.g., cobicistat) or lack of clinical information. Currently, 2DRs cannot be recommended during pregnancy or breastfeeding [87].

In addition, hepatitis B virus (HBV) chronic infection has been an exclusion criterion for all 2DR clinical trials. HBV and HIV coinfection is common, especially in certain low-income regions [88]. Some antiretroviral drugs used for the treatment of HIV infection are also active against HBV, such as 3TC, FTC, TDF and TAF. Previous research discourages the use of 3TC monotherapy in HIV–HBV coinfected patients, due to a high risk of the emergence of 3TC resistance mutations [89]. Therefore, 2DRs cannot be recommended in patients with HBV and HIV co-infection.

Two-drug regimens cannot be recommended in severely immunosuppressed patients (i.e., CD4 lymphocyte count under 200 cells/mm^3^) due to the lack of evidence for these patients. Future studies with second-generation INSTIs (e.g., DTG) are needed in severe immunodepression settings.

Focusing on high VL (i.e., VL above 500,000 copies/mL), 2DRs cannot be routinely recommended due to previous results. Most clinical guidelines do not recommend starting a 2DR in these patients due to the lack of clinical data at this time. Additional clinical trials designed to evaluate 2DR efficacy in the context of high VL are needed.

## 3. Conclusions

The clinical experience with 2DRs for the treatment of HIV infection has greatly increased during the last 20 years, showing comparable efficacy, safety and convenience to 3DR, both in ART-naïve and treatment-experienced, virologically suppressed patients. The results of several clinical trials have driven 2DR to the top of many HIV clinical guidelines, progressing from an alternative regimen in difficult-to-treat patients to first-line HIV treatment in treatment-naïve as well as in virologically suppressed patients. Moreover, development of new 2DRs continues, with new presentations that are even more convenient being sought. With the exceptions of a few groups of patients (i.e., those who are pregnant, severely immunosuppressed or with chronic HBV coinfection), 2DRs have become a suitable option for most HIV-infected patients.

## Figures and Tables

**Figure 1 microorganisms-10-00433-f001:**
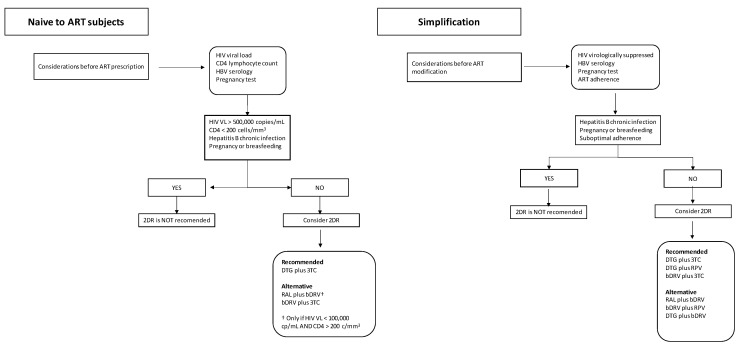
Proposed algorithm for two-drug ART prescription.

**Table 1 microorganisms-10-00433-t001:** Position of two-drug regimens in HIV clinical guidelines.

**Naïve-to-ART Patients**				
	**GeSIDA**	**EACS**	**DHHS**	**Observations**
DTG + 3TC	Recommended	Recommended	Recommended	HbS Ag-negative HIV VL < 500,000 copies/mL
RAL + bDRV	Not recommended	Alternative	Alternative	CD 4 count > 200 cells/mm^3^ HIV VL < 100,000 copies/mL
bDRV + 3TC	Not recommended	Not recommended	Alternative †	
**Simplification in Virologically Suppressed Patients**				
	**GeSIDA**	**EACS**	**DHHS**	**Observations**
DTG + RPV	Recommended	Recommended	Recommended	
DTG + 3TC	Recommended	Recommended	Recommended	
bPI + 3TC	Alternative	Recommended	Alternative ‡	‡ DRV is preferred over LPV and ATV
DTG + bDRV	Recommended	Alternative	Alternative	
bDRV + RPV	Not recommended	Alternative	Not recommended	

ART: anti-retroviral treatment; GeSIDA: Grupo de estudio del SIDA; EACS: European AIDS Clinical Society; DHHS: Department of Health and Human Services; HbS Ag: hepatitis B surface antigen; HIV VL: human immunodeficiency virus viral load; DTG: dolutegravir; 3TC: lamivudine; RAL: raltegravir; bDRV: boosted-darunavir; RPV: rilpivirine; bPI: boosted-protease inhibitor; DRV: davunavir; LPV: lopinavir; ATV: atazanavir. † If chronic kidney disease is present, and only DRV/r.

**Table 2 microorganisms-10-00433-t002:** Main clinical trials comparing two- versus three-drug regimens for HIV infection.

Clinical Trial	2DR Arm	Comparator	Subject Population	Sample Size	Follow-Up	HIV-RNA ≤ 50 cp/mL, Absolute Risk Difference (95% CI)	Virological Response in 2DR Arm vs. Comparator
GARDEL	LPV/r + 3TC	LPV/r + 2 NRTIs	Naive to ART	214 vs. 202	48 weeks	4.6 (–2.2 to 11.8) †	88.3% vs. 83.7% †
OLE	LPV/r + 3TC	LPV/r + 2 NRTIs	Virologically suppressed	118 vs. 121	48 weeks	1.19 (–7.10 to 9.50) †	88.0% vs. 87.0% †
SALT	ATV/r + 3TC	ATV/r + 2 NRTIs	Virologically suppressed	133 vs. 134	96 weeks	1.39 (–8.50 to 11.30) ‡	69.9% vs. 71.3% ‡
ATLAS-M	ATV/r + 3TC	ATV/r + 2 NRTIs	Virologically suppressed	133 vs. 133	48 weeks	6.77 (–2.20 to 15.70) *	89.5% vs. 79.7% *
DUAL-GESIDA 8014	DRV/r + 3TC	DRV/r + 2 NRTIs	Virologically suppressed	126 vs. 123	48 weeks	–3.79 (–10.90 to 3.30) *	88.9% vs. 92.7% *
SECOND-LINE	LPV/r + RAL	LPV/r + 2 or 3 NRTIs	First-line virological failure	270 vs. 271	48 weeks	1.8 (–4.7 to 8.3) ¶	80.8% vs. 82.6% ¶
SELECT	LPV/r + RAL	LPV/r + 2 or 3 NRTIs	First-line virological failure	260 vs. 255	48 weeks	3.4 (–8.4 to 1.5) ¶	89.7% vs. 87.6% ¶
EARNEST	LPV/r + RAL	LPV/r + 2 or 3 NRTIs	First-line virological failure	433 vs. 426	96 weeks	–0.1 (–5.0 to 4.8) §	64.0% vs. 60.0% §
GEMINI 1 and 2	DTG + 3TC	DTG + FTC/TDF	Naive to ART	719 vs. 722	48 weeks	–1.7 (–4.4 to 1.1) *	91.0% vs. 93.0% *
TANGO	DTG + 3TC	TAF-based 3DR	Virologically suppressed	369 vs. 372	48 weeks	–0.3 (–1.2 to 0.7) *	93.2% vs. 93.0% *
SWORD 1 and 2	DTG + RPV	3DR	Virologically suppressed	516 vs. 512	48 weeks	–0.2 (–3.0 to 2.5) *	95.0% vs. 95.0% *
DUALIS	DTG + bDRV	DRV-based 3DR	Virologically suppressed	131 vs. 132	48 weeks	–1.6 (–9.9 to 6.7) *	86.3% vs. 87.9% *
NEAT001/ANRS 143	RAL + DRV/r	DRV/r + FTC/TDF	Naive to ART	401 vs. 404	123 weeks	4.0 (–0.8 to 8.8) ††	87.6 % vs. 89.7% ††
PROBE-2	bDRV + RPV	3DR	Pre-treated	80 vs. 80	24 weeks	–3.75 (–11.63 to 5.63) *	90.0% vs. 93.8% *
FLAIR	CAB + RPV	DTG/3TC/ABC	Pre-treated	283 vs. 283	48 weeks	0.4 (–3.7 to 4.5) *	93.6% vs. 93.3% *

2DR: two-drug regimen; LPV/r: ritonavir-boosted lopinavir; 3TC: lamivudine; NRTIs: nucleoside/nucleotide reverse transcriptase inhibitors; ATV/r: ritonavir-boosted atazanavir; DRV/r: ritonavir-boosted darunavir; RAL: raltegravir; FTC/TDF: emtricitabine/tenofovir-disoproxil-fumarate; DTG: dolutegravir; TAF: tenofovir alafenamide; 3DR: three-drug regimen; RPV: rilpivirine; bDRV: boosted-darunavir; CAB: cabotegravir; DTG/3TC/ABC: dolutegravir/abacavir/lamivudine. † Intention-to-treat, exposed, snapshot; ‡ Time to loss of virological response (TLOVR); * US Food and Drug administration (FDA) snapshot algorithm; ¶ Custom analysis equivalent to FDA snapshot algorithm; § Custom composite end-point; †† Kaplan–Meier estimated proportions analysis.

**Table 3 microorganisms-10-00433-t003:** Adverse events and discontinuation rates within clinical trials comparing 2DRs and 3DRs.

Clinical Trial	2DR Arm	Comparator	Total Number of Patients with One or More AEs	Total Number of Patients with One or More SAEs	Discontinuation Because of Adverse Events or Death	eGFR Difference in mL/min/1.7 m^2^
GARDEL	LPV/r + 3TC	LPV/r + 2 NRTIs	65 (30%) vs. 88 (44%) †	1 (<1%) vs. 0 ‡	3 (1%) vs. 16 (8%)	Not reported
OLE	LPV/r + 3TC	LPV/r + 2 NRTIs	63 (53%) vs. 70 (58%)	5 (4%) vs. 8 (7%)	1 (1%) vs. 4 (3%)	Not reported
SALT	ATV/r + 3TC	ATV/r + 2 NRTIs	99 (70.7%) vs. 99 (70.2%)	10 (7.5%) vs. 9 (6.7%)	7 (5.3%) vs. 11 (8.2%)	−0.8 vs. −1.4
ATLAS-M	ATV/r + 3TC	ATV/r + 2 NRTIs	33 (24.8%) vs. 40 (30.1%	3 (2.3%) vs. 4 (3.0%)	4 (3.0%) vs. 8 (6%)	+2 vs. −5
DUAL-GESIDA 8014	DRV/r + 3TC	DRV/r + 2 NRTIs	88 (70%) vs. 93 (76%)	6 (5%) vs. 6 (5%)	1 (1%) vs. 2 (2%)	Not reported
SECOND LINE	LPV/r + RAL	LPV/r + 2 or 3 NRTIs	993 vs. 895 *	23 (8.5%) vs. 24 (8.9%)	11 (4%) vs. 8 (3%)	−5.2 vs. −4.7
SELECT	LPV/r + RAL	LPV/r + 2 or 3 NRTIs	18 (7%) vs. 27 (11%)	5 (1.9%) vs. 5 (1.9%)	3 (1.2%) vs. 3 (1.2%)	Not reported
EARNEST	LPV/r + RAL	LPV/r + 2 or 3 NRTIs	104 (24%) vs. 92 (23%)	93 (22%) vs. 91 (21%)	30 (6.9%) vs. 30 (7.0%) ¶	−5.4 vs. −11.2
GEMINI 1 and 2	DTG + 3TC	DTG + FTC/TDF	543 (76%) vs. 579 (81%)	50 (7%) vs. 55 (85)	10 (1%) vs. 13 (2%)	−2.1 vs. −15.5
TANGO	DTG + 3TC	TAF-based 3DR	295 (79.9%) vs. 292 (78.7%)	21 (5.7%) vs. 16 (4.3%)	13 (3.5%) vs. 2 (0.5%)	−7.7 vs. −3.0
SWORD 1 and 2	DTG + RPV	3DR	395 (77%) vs. 364 (71%)	27 (5%) vs. 21 (4%)	17 (3%) vs. 3 (1%)	Not reported
DUALIS	DTG + bDRV	DRV-based 3DR	104 (78.2%) vs. 100 (75.2%)	7 (5.3%) vs. 7 (5.3%)	14 (4.6%) vs. 5 (1.6%)	Not reported
NEAT001/ANRS 143	DRV/r + RAL	DRV/r + FTC/TDF	34 vs. 38 **	73 (18.2%) vs. 61 (15.1%)	1.5% vs. 2.6%	+0.8 vs. −4.6
PROBE-2	bDRV + RPV	3DR	6 (7.5%) vs. 3 (3.4%)	Not reported	6 vs. 0	Not reported
FLAIR	CAB + RPV	DTG/3TC/ABC	267 (94%) vs. 225 (80%)	18 (6%) vs. 12 (4%)	9 (3%) vs. 4 (1%)	Not reported

† Grade 2–3 AEs; ‡ Only drug-related SAEs; * Total number of AEs; ¶ Only discontinuations due to death; ** Only number of AEs leading to treatment modification.

## Data Availability

No new data were created or analyzed in this study. Data sharing is not applicable to this article.

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
