# Peer review of "Two-Drug Regimens for HIV—Current Evidence, Research Gaps and Future Challenges"

_microorganisms, 2022, doi:10.3390/microorganisms10020433_

Round 1

Reviewer 1 Report

Microorganisms

 MDPI

Review

Two-drug-regimens for HIV. Current evidence, research gaps 2 and future challenges. 3

Alexandre Pérez-González MD1,2, Inés Suárez-García MD3, Hortensia Álvarez MD4, Antonio Ocampo PhD2 and 4 Eva Poveda PhD1

The study Two-drug-regimens for HIV. Current evidence, research gaps 2 and future challenges is a review research based on the most relevant clinical trials focused on 2DR, that is a new paradigm among the HIV treatment strategies.

The comparative results of the trials with 3DR and 2DR are presented realistically and convincible for the advantages of 2DR, balancing with the discussions on the limits and unclarified aspects of this strategy.

There are well pointed the two recommendations of 2DR as first line treatment in naïve patients and as simplification therapy in antiretrovirals experienced patients with HIV controlled infection.

Comments to authors: 

The article is well written, and I approve the publication.
Although it is optional, in order to be very accurate, I suggest introducing a supplementary material for explaining the screening process of references, according to PRISMA flow diagram.

Reviewer 2 Report

Major comments:

The argument for 2DRs has always been the reduction of toxicity. However, only scarce data support this argument following treatment switch. In this review, the authors report similar AE rates between 2DR and 3DR. With this in mind, they should add a second table that would detail reported AEs (by categories and those leading to withdrawal) and notable clinical differences between arms for each of the trials listed in Table 1. This would allow a much more straightforward comparison of 2DR vs 3DR. For the AE, I do not suggest that the authors list all AEs, but they should list numbers per AE grade, for example (or percentages). For clinical changes, I believe that at least renal, lipids markers, and weight should be included in this new table (the authors should feel free to add other criteria).

In addition to the first point, the authors should discuss the few trials that have shown partial reversal of pressure on kidney function for patients who removed TDF from their regimens (regardless of that happening in the context of 2DR or not). And they should also mention that, reciprocally, there is no data showing reversion of potential cardiac link with the withdrawal of ABC. This should be reviewed and detailed clearly. Although I personally believe, like the authors, that 2DR may be generally less toxic than 3DR, this review should not be about what we believe but rather about demonstrable facts. It should also be noted (for example, in the conclusion) that 2DRs have not yet been tested against BIC/TAF/FTC (Biktarvy). Please discuss these points/other potential arguments against 2DRs in the document.

In the abstract, it is unfair to state that “3DRs… is associated to several side effects etc…”. I suggest using “can be associated with”. In clinical practice, many patients support 3DR with little side effects and minor toxicity (in contrast to “important ART-related toxicity”).

Abstract: I wouldn’t say that 2DR are “at the top” of many HIV guidelines. For example, as far as initiation, only dolutegravir plus lamivudine is part of the European guidelines with a caveat regarding viral loads. Please edit.

Figure 1: in the proposed algorithm for 2DR prescription, the authors miss what is arguably the most important clinical consideration for simplification: adherence to treatment. This characteristic must be included in the “Simplification” arm.

Should the authors add a “Methods” section where they describe what bibliographic terms they used and what sources they explored (e.g., PubMed, conference proceedings)? (Please see with editorial instructions if this suggestion is appropriate or not). The authors should also clarify what criteria they applied to define “main clinical trials”. For example, did they use a cut-off on enrollment/participation number?

Table 1: the authors need to report the number of participants for each study/arm. This is important to assess the relevance of the results. Virological response is reported, but it should be specified whether those are by FDA snapshot analysis, ITT, or PP (in the table).

Line 158: the authors should include more information about reasons for treatment simplification beyond toxicity (pill burden, ddi, etc). In addition, simply removing ABC or TDF but at the same time adding a boosted PI, for example, is arguably not a straightforward option to reduce toxicity. The authors should temper the second sentence and introduce some nuance and details to this paragraph.

Line 239: it is inaccurate to state that there is persistent replication in lymphatic tissues under ART. In most patients, it has been shown through phylogenetic studies, for example, that there is no persistent replication after cART has been initiated (see work from Mary Kearney and colleagues and many other papers). The authors should correct or remove this sentence—same line 267.

In the paragraph about long-acting injectables, the authors should include recent data showing the feasibility to skip the oral induction phase of CBG/RPV (and its equivalence with oral induction).

Line 367: the authors identify “immune activation” as a remaining question. Some of it has been answered in GEMINI. In addition, in the following sentence, the authors discuss not immune activation but concerns about the reservoirs. They should clarify what the concerns are, and in which category they fall (i.e., which ones are research-based vs. clinically relevant). As far as the sentence starting line 374, the authors express concerns about inflammatory biomarkers (should it be clinical outcomes?), but the end of the sentence is about long-term “efficacy”. This entire paragraph should be edited to segregate the topics (i.e., efficacy, reservoirs, inflammation, etc…).

Minor comments:

Abstract: antiretroviral, please edit

Abstract: the authors may want to change the term “development” since there was a period of 2DR before 1996.

Line 59: “On the other hand” or equivalent instead of “In addition” since the authors now switch to examining tenofovir’s toxicity.

Line 84: the sentence is incomplete and should be edited to include “clinical evidence that supports the use of 2DRs…”. What I mean is that there could be no “clinical evidence of 2DR”. Please edit.

Table 2: “Simplification…” should be in bold.

Minor consideration: the authors typically invert “bDRV” when it is used with RAL vs. RPV. Is it an indication that they consider RAL to be the anchor drug in the regimen RAL+b/DRV? As far as I know, there have been no RCTs comparing RAL to RPV head-to-head, so it is hard to argue that boosted DRV is not the anchor component in both regiments (i.e., with RAL and RPV). It may be appropriate to re-organize and harmonize the listing of these various drugs in the text and tables/figures (e.g., Table

In the integrase paragraph starting line 129, the authors did not discuss RAL-based trials, including EARNEST. It could be important to include in the text given its clinical settings.

Paragraph starting line 129, the authors should clarify what the differences were between TANGO and SWORD.

Line 172, it should be noted that the impact of these minor changes on clinical outcomes remain unknown.

Line 210: although it is true that no protocol-defined virological failure with resistance was reported for GEMINI-1 and 2, there was one case of resistance that did not match the definition of PDVF. It is thus untrue to state that there was no case of resistance and this specific case should be mentioned in the review.

Line 246 “at controlling”

Line 292: the convention is not to use the word “subjects”; “participants”, “people” or “individuals” can be used. Same line 302.

Line 297: GEMINI “are the most important clinical trials”? This sounds like a judgement of value on all other 2DR RCTs that the authors have reviewed. If the authors mean that GEMINI are the largest RCT by number of participants, they should state it clearly (they will have to double check vs. EARNEST for example). If they mean that they believe that GEMINI are indeed the most important trials in a specific way they should temper the sentence with “It is our opinion that…” and clarify why they believe these trials are so important. Please edit.

Line 333: it’s not the drugs that are subcutaneous or intramuscular; antiretroviral drugs are formulated for subQ or IM injections. Please edit the sentence.
